# Discovery and Validation of Clinically Relevant Long Non-Coding RNAs in Colorectal Cancer

**DOI:** 10.3390/cancers14163866

**Published:** 2022-08-10

**Authors:** Madison Snyder, Susana Iraola-Guzmán, Ester Saus, Toni Gabaldón

**Affiliations:** 1Barcelona Supercomputing Centre (BSC-CNS), Plaça Eusebi Güell, 1-3, 08034 Barcelona, Spain; 2Institute for Research in Biomedicine (IRB Barcelona), The Barcelona Institute of Science and Technology, Baldiri Reixac, 10, 08028 Barcelona, Spain; 3Catalan Institution for Research and Advanced Studies (ICREA), 08010 Barcelona, Spain; 4Centro de Investigación Biomédica En Red de Enfermedades Infecciosas (CIBERINFEC), 08028 Barcelona, Spain

**Keywords:** colorectal cancer, lncRNAs, biomarkers

## Abstract

**Simple Summary:**

Recent efforts in biomedical research have focused on the identification of molecular biomarkers to improve the diagnosis, prognosis and eventually treatment of the most common human diseases worldwide, including cancer. In this context, a large number of studies point to a pivotal role of long non-coding RNAs (lncRNAs) in the pathophysiology of carcinogenesis, suggesting diagnostic or therapeutic potential. However, for most of them, supporting evidence is scarce and often based on a single large-scale analysis. Here, focusing on colorectal cancer (CRC), we present an overview of the main approaches for discovering and validating lncRNA candidate molecules, and provide a curated list of the most promising lncRNAs associated with this malignancy.

**Abstract:**

Colorectal cancer (CRC) is the third most prevalent cancer worldwide, with nearly two million newly diagnosed cases each year. The survival of patients with CRC greatly depends on the cancer stage at the time of diagnosis, with worse prognosis for more advanced cases. Consequently, considerable effort has been directed towards improving population screening programs for early diagnosis and identifying prognostic markers that can better inform treatment strategies. In recent years, long non-coding RNAs (lncRNAs) have been recognized as promising molecules, with diagnostic and prognostic potential in many cancers, including CRC. Although large-scale genome and transcriptome sequencing surveys have identified many lncRNAs that are altered in CRC, most of their roles in disease onset and progression remain poorly understood. Here, we critically review the variety of detection methods and types of supporting evidence for the involvement of lncRNAs in CRC. In addition, we provide a reference catalog that features the most clinically relevant lncRNAs in CRC. These lncRNAs were selected based on recent studies sorted by stringent criteria for both supporting experimental evidence and reproducibility.

## 1. Introduction

Cancer is among the most prominent life-threatening diseases in the world, with over 19 million newly diagnosed cases in 2020 alone [1]. Notably, colorectal cancer (CRC) is one of the most commonly diagnosed cancers, with nearly 2 million new cases each year (~10% of all new cancer cases). Moreover, CRC is the second leading cause of all cancer-related deaths, claiming almost 1 million lives in 2020 [1]. Although extremely deadly in the advanced stages, the development of CRC is gradual. Beginning from the pathological transformation of normal colonic epithelium to adenomatous polyp, CRC ultimately leads to invasive cancer [2]. CRC progression is generally categorized into five stages (0 to IV), depending on the extensiveness and clinical features [3]. The lethality of CRC is largely correlated to the stage of the disease at diagnosis. At its early stages (stages 0–II), CRC is very treatable, with 5-year survival rates as high as 90%. However, only 38% of CRC cases are diagnosed at an early, localized stage. By the later stages, the 5-year survival rate dramatically decreases to as low as 14% [4]. Hence, early diagnosis of CRC is key to saving lives.

While CRC has historically affected older populations, recent trends show an increase in cases in those under 50 years old. This has resulted in a decrease in the median age of patients with CRC from 72 to 66 years old. This is particularly concerning, as early onset CRC is often diagnosed at advanced (less treatable) stages, as compared to CRC in the traditional patient population [5]. Despite these negative trends, overall CRC mortality and incidence rates have consistently improved each year, reflecting the rise in preventative screenings, new testing, and targeted treatments [4].

Given the prominence and severity of this disease, there has been a large effort through recent research to better understand its causes, prognosis, and outcome. However, further research is still needed to improve prevention and treatment through more novel discoveries. This includes the identification of diagnostic and prognostic biomarkers. Compared to traditional diagnostic methods, such as colonoscopy or prognostic methods, including measuring tumor size and metastasis, the detection of biomarkers is less invasive. Unlike traditional tests, biomarker analysis can be carried out using urine, fecal, plasma, saliva or serum samples [6]. Thus, biomarkers have the potential to distinguish between benign and cancerous tumors (polyps vs. carcinomas) less invasively, while providing more accurate predictions of disease progression, likelihood of relapse, and even chance of onset [6].

Biomarkers have been a tremendous success at better informing diagnosis, prognosis, treatment, and preventative measures in other cancers [7,8,9,10]. One of the most famous examples, mutations in *BRCA1/BRCA2 genes*, have given patients the ability to reliably assess their risk of developing breast cancer in their lifetime, as well as their risk of relapse after a first bout of breast cancer. Additionally, mutations in *BRCA1/BRCA2* genes can be used for informing patient care and treatment strategies after disease onset [7]. Similarly, the diagnostic lncRNA biomarker, *PCA3*, has been approved for clinical use in suspected cases of prostate cancer [11,12]. The ratio of *PCA3*, which encodes a prostate-specific RNA, and a prostate-specific antigen (PSA), is measured in urine samples and used to increase the specificity of the diagnosis [11,12,13]. In CRC, many different molecules have been identified as potential biomarkers, including lncRNAs. LncRNAs are RNA molecules longer than 200 bp that do not code for proteins. They have been broadly classified into sense, antisense, bidirectional, intronic and intergenic lncRNAs, depending on their relative position to protein-coding genes [14]. The vast majority of characterized lncRNAs are synthesized by Polymerase II, and subsequently spliced and 5′-capped. Additionally, some lncRNAs are also polyadenylated [15]. lncRNAs are poorly conserved, showing fewer exons, and generally have limited expression. Many lncRNAs are localized in the cell nucleus, where they exert regulatory functions by binding to DNA or DNA-associated proteins [16]. Other lncRNAs are transported to the cytosol, where they can interact with other cytosolic molecules. LncRNA mechanisms of action are generally classified into the following four main groups: chromatin regulation, gene regulation, scaffolding and condensation, and post-transcriptional regulation, as illustrated in Figure 1 [17]. LncRNA expression is generally limited by space and time, as it is often tissue or cell-type specific. Alterations in the pattern of expression of lncRNAs have been recurrently reported in cancer, where they can act as either oncogenes or tumor suppressors [18]. Overexpression and/or downregulation of lncRNAs in tumors is often associated with additional epigenetic alterations, such as DNA (de)methylation of promoters or enhancers [19,20,21]. It has been shown that differential expression of a subset of lncRNAs is associated with CRC heterogeneous features and also with functional pathways that mediate CRC, such as TGF-β and WNT pathways, immunity, epithelial-mesenchymal-transition (EMT), and angiogenesis [22]. The biomarker potential of lncRNAs has been increasingly studied in CRC in recent years [23]. However, most of these candidate lncRNAs currently lack proper experimental validation or characterization to be considered promising targets.

In this review, we outline the different modes of discovery for identifying lncRNAs as potential biomarkers in CRC and the advantages and disadvantages of each of these methods. We also discuss the common methods for evaluating the role and prognostic potential of previously identified lncRNAs. Based on recent studies, we provide a list of the most promising candidates based on study reproducibility and level of experimental characterization. Finally, we discuss the current use of lncRNAs as biomarkers in CRC and their potential as therapeutic targets in the future. Figure 2 provides an overview of the discovery and validation approaches for identifying clinically relevant lncRNAs, as described in detail in this review.

## 2. Approaches to Identify Relevant lncRNAs in CRC

Traditionally, approaches for determining the diagnosis and prognosis of CRC cases have been limited to non-molecular factors. In regard to diagnosis, colonoscopy and subsequent biopsy have been the gold standard in CRC screening [24]. Similarly, prognostic tools have been dominated by clinical and histological criteria including measurements of tumor size, tumor grade (stages described previously), and patient age among others [25]. However, recent research has started to move away from these approaches. Instead, focus has shifted towards identifying the prognostic and therapeutic potential of molecular biomarkers, including lncRNAs. LncRNAs comprise the majority of noncoding RNAs, many of which have unknown functions [14,26]. With the rise of next-generation sequencing (NGS) technologies and the subsequent ability to collect and analyze large volumes of data, many lncRNAs with prognostic and therapeutic potential in CRC have been identified. However, many of these candidates result from large-scale approaches that do not constitute conclusive proof, and therefore require further validation. Here, we provide a summary of the most used methods for identifying lncRNAs involved in CRC and discuss the advantages and disadvantages of each.

### 2.1. RNA Sequencing

With the increased accessibility of NGS, many researchers have begun to study transcriptional alterations in cancer through RNA sequencing (RNA-Seq). This technique allows researchers to reconstruct and quantify the expression of transcripts present in biological samples [27]. RNA-Seq studies that compare a CRC tumor and healthy tissue from the same patient can be used to uncover the differential expression and somatic mutations of various lncRNAs. These differentially expressed or mutated lncRNAs have the potential to be involved in the onset and progression of CRC, warranting further study.

Due to its untargeted approach, RNA-Seq uniquely allows for the discovery of novel lncRNAs. However, RNA-Seq does have some disadvantages. One significant challenge in detecting lncRNAs through RNA-Seq is their low relative abundance. Compared to protein coding genes, lncRNAs are extremely lowly expressed, constituting a minute fraction of the total RNA transcripts in a sample. To address this issue, target enrichment techniques utilizing probe-based strategies have been developed, enabling more effective lncRNA detection [28,29]. Given the large amounts of data that RNA-Seq analyses produce, there is also a risk of false positive detection of transcripts. This can be due to noisy expression or from transcripts that encode for proteins [27]. In fact, one study that investigated the reproducibility of differential expression results from identical replicates found that up to 8% of differentially expressed (DE) genes identified by RNA-Seq were false positives, even when using stringent identification parameters [30]. Maybe most concerningly, some studies have questioned the reproducibility of RNA-Seq results, as the resulting analyses are often dependent on the quality control, alignment, and quantification tools that are used in the analysis pipeline [31,32]. Moreover, the lack of normalization in analytical statistical methods often means that results are over dispersed and replicate dependent [32,33].

### 2.2. Microarrays

Microarrays are another genome-wide screening approach that can be used to identify lncRNAs involved in CRC. Microarrays are glass slides, lined with selected DNA oligonucleotide sequences in known locations that can hybridize specific lncRNAs (converted to cDNA) from a biological sample. Complementary base pairing between the sample and the oligonucleotide sequences on the chip produces light proportional to gene expression. Thus, hybridization allows for the detection of gene expression changes in a previously selected group of candidate lncRNAs in cancer cells [34,35]. Compared to RNA-Seq, microarrays are often less costly and require less complex and extensive bioinformatic analysis. However, they do require a preselection of lncRNAs that are suspected to have a biologically relevant function in CRC. Consequently, microarrays do not allow for the discovery of novel lncRNAs [27]. Recent work has also called into question the reliability and reproducibility of microarrays, due to unstable surface deposition chemistries [36]. A comparison of five different microarray data platforms revealed that there is poor concordance between systems in their output of results [37,38]. Additionally, poor sensitivity in detecting lowly expressed molecules, such as lncRNAs, limits detection efficiency [39].

### 2.3. CRISPR-Cas9 Screening

CRISPR-Cas9 screening for lncRNAs is one of the most optimizable modes of discovery for identifying candidate lncRNAs because of the many variables that can be altered when building a screen [40,41]. CRISPR-Cas9 screens can be designed with cells perturbed through loss of function techniques (inhibition or deletion) or gain of function techniques (activation). Screenings can also be specific to a number of lncRNA targets or to larger lncRNA pools. After selecting a perturbation method and the target lncRNAs, a single-guide RNA (sgRNA) library must be selected. NGS data of sgRNA counts are then used to identify lncRNAs associated with diseases such as CRC [42,43]. Compared to RNA-Seq or microarray analyses, there is no need to assume that differential expression implies function. Instead, CRISPR-Cas9 screening in a variety of cell lines can be used to identify lncRNAs through cancerous phenotypes, such as proliferation or drug resistance, independent of differential expression [42]. A major benefit of CRISPR-Cas9 screening is its customizability and effective targeting of lncRNAs [44]. However, statistical analyses of these screens are often hindered by the use of limited replicates [43]. Given the recent implementation of this technique, there is also a need to establish benchmarks to improve evaluation and reproducibility [42].

### 2.4. Bioinformatic Approaches

Several bioinformatic approaches that analyze genomic or transcriptomic data can be used to identify lncRNAs with prognostic or therapeutic potential in CRC. One such approach is the “detecting lncRNA cancer association” (DRACA) method, which was developed to evaluate potential molecular biomarkers by predicting lncRNA-cancer associations [45,46]. DRACA uses matrix factorization to consider interactions between lncRNAs, cancer prognosis, and other factors, such as miRNAs and genes, to predict lncRNA-cancer association. Using already available data on cancer prognosis, this approach results in both novel and biologically meaningful discoveries [46]. Likewise, the tool OncodriveFML was developed to identify known somatic mutations in genomic elements (such as lncRNAs) to predict those that have undergone positive selection during tumorigenesis. These lncRNAs have a high functional mutation bias and further role in CRC [47]. While innovative in its approach, OncodriveFML does have some limitations. OncodriveFML relies on the characterization of the functional impact of mutations in its calculations, meaning less characterized mutations in noncoding regions will not be counted. It also only predicts based on nucleotide substitutions, forgoing the identification of lncRNAs with insertions or deletions [47]. Another tool, the ExInAtor, also relies on mutational patterns of tumoral DNA, rather than changes in gene expression, to identify tumor driver lncRNAs [48]. This approach is especially advantageous for its specificity, rapid computation, and its ability to identify lncRNAs involved in tumorigenesis and not solely in upstream regulatory processes. However, ExInAtor does not evaluate the functional impact of mutations in lncRNAs, limiting the sensitivity of this approach and leaving room for many false negatives. Candidate lncRNAs identified by this approach also unexpectedly harbored many repeats and had lower GC content, indicating a possible bias of the tool [48]. An additional source is the RNA Atlas expanded with non-coding RNAs, which covers more than 3310 novel lncRNAs from RNAseq experiments that were performed in more than 300 human tissues and cell lines, including CRC [49].

## 3. Experimental Validation of Candidate lncRNAs

The discovery techniques outlined in the previous section are useful approaches for identifying lncRNAs of interest. However, given the existing biases and limitations of these approaches, further evaluation is needed to assess the true functional significance and clinical potential of these candidate lncRNAs. Here, we examine several common approaches used for this purpose.

### 3.1. Expression Profiling

As discussed above, one of the most common ways to identify lncRNAs with prognostic potential in CRC is through expression profiling in tumor and healthy tissues using RNA-Seq or microarrays. However, inconsistencies in the analysis of gene expression data can limit the reliability of these results. Real-time quantitative PCR (RT-qPCR) is the gold standard for verifying credible gene expression results [50]. In regard to lncRNAs in CRC, RT-qPCR can be used to amplify a target lncRNA in both tumoral and normal tissue. After each amplification cycle, the quantity of amplified target lncRNA is directly measured. Quantification assessments are then compared between tissue types to generate an expression profile [51]. While being a trusted validation method, many variables may jeopardize the reliable implementation of RT-qPCR. These include the quality of RNA, quantity and quality of reference genes, type of priming approach, and target abundance [52,53]. Although RT-qPCR is limited by technical and biological variability, it is often viewed as the benchmark technology in expression profiling, due to its sensitivity and specificity [50,51,52,53].

Another approach for validating expression profile data is through fluorescence in situ hybridization (FISH). FISH combines microscopy with the use of fluorescent probes that bind a target nucleic acid sequence to measure its presence and abundance within a particular tissue [54]. FISH can monitor RNA targets with high specificity and sensitivity and can accommodate different tissue types, including formalin-fixed paraffin embedded (FFPE) tissues, in which CRC tissues are often stored [55,56]. While a promising method, FISH is highly dependent on probe design, leaving room for variability in efficiency and accuracy of the approach [55]. An alternative methodology is RNAScope, which is a combined technique for the detection of target molecules of RNA in FFPE tissues [57]. In brief, multiple oligonucleotide-probes hybridize to target RNA molecules, then labeled (enzymatically or with fluorophores) amplifier molecules are hybridized to the probes, allowing detection by microscopy. Compared with previous in-situ hybridization techniques (ISH), RNAScope presents different advantages, such as increased sensitivity and specificity, and the possibility of multiple target detection. For this reason, it has been proposed as a valid alternative to RT-qPCR, for the characterization of biomarkers in cancer tissues [58,59,60,61]. More recently, the increasing interest in the localization and quantification of lncRNAs has stimulated the application of RNAScope to the detection of lncRNAs in FFPE tissue of patients with cancer [62,63]. The time-consuming set-up and the elevated cost are the main drawbacks of this methodology.

### 3.2. Clinical Significance

To evaluate the clinical significance of candidate lncRNAs, it is important to consider the clinicopathological, diagnostic, and prognostic value of these molecules. The correlation between candidate lncRNA expression and clinicopathological characteristics, such as age, sex, tumor size, tumor grade, and distant metastasis, is often analyzed to provide preliminary evidence of a lncRNA’s clinical relevance in CRC [64,65]. The most relevant analytical methods for defining clinical significance are summarized in Figure 3, and further explained here (Figure 3).

In terms of diagnosis, the receiver operating characteristics curve (ROC) is a trusted method to evaluate the sensitivity and specificity of biomarkers in CRC. ROC curves plot the true positive rate against the false positive rate for discriminating between CRC/non-CRC cases [66]. Consequently, ROC analysis can be used to determine the diagnostic relevance of a lncRNA biomarker, with the area under the ROC curve (AUC) as the quantifiable measurement of diagnostic accuracy. An ROC curve with an AUC of 0.5 describes a lncRNA biomarker with no discriminating ability between the CRC and non-CRC cases. AUCs above 0.5 provide some discriminatory ability, with an AUC of 1 representing a perfect biomarker [66].

Defining the prognostic value of lncRNAs is also essential to improve disease projections and patient care through more personalized treatment options. One common way to evaluate the prognostic potential of lncRNAs is through Cox regression analysis. Cox regression analysis is used to determine the relationship between predictors and the time that passes before an event occurs [67]. In this case, univariate Cox regression models are used to define the relationship between the predictor (lncRNA expression levels) and patient survival time. Univariate Cox regression between lncRNAs and overall survival has been carried out in multiple cancers, including colorectal cancer [68,69]. Multiple regression analysis can be used to analyze the significance between multiple variables, painting a more comprehensive picture of the prognostic potential of lncRNAs. This method can be more sensitive to comorbidities or covariates that potentially affect patient prognosis [70]. In this case, multiple lncRNA expression profiles, along with other variables, such as patient age, histology, or tumor grade, can be considered, giving a more advanced prognostic evaluation of the lncRNA signature.

Additionally, the Kaplan–Meier method is a powerful tool used to visualize survival differences in CRC. Kaplan–Meier plots show the overall survival rate of patients over a designated period of time. By separating patients into groups of high and low expression of a particular lncRNA, these plots can be used to visualize the effect of lncRNA expression on overall survival in CRC. Unlike Cox regression analyses, multiple variables cannot be considered in this survival analysis.

Given that a patient’s prognosis is a strong determinant of the therapeutic action that follows, prognostic and clinicopathological testing is incredibly important in defining the clinical significance of lncRNAs [69,71,72].

### 3.3. Regulatory Significance

In many cases, previously identified lncRNAs are predicted to have roles in tumorigenesis, migration, and invasion of cancer cells. To study the effect of aberrantly expressed lncRNAs on cancer development, researchers have investigated lncRNA-dependent progression and molecular networking of cancer cells through a variety of methods. In regard to cancer progression, dysregulating (silencing or overexpressing) a lncRNA of interest allows for the study of lncRNA-dependent cell proliferation, migration, invasion, and colony formation. The effect of an aberrantly expressed lncRNA on these essential cancer processes can be tested through apoptotic/CCK-8, wound healing, transwell, and cell formation assays, respectively [71]. These regulatory assays can be performed in several models of CRC, including CRC cell lines, colon organoids, and mouse models.

The regulatory significance of target lncRNAs is also highly dependent on their binding partners and network interactions. Proteins and mRNAs that are coexpressed with lncRNA targets can be predicted through bioinformatic softwares, including RPIseq, lncPRO, lncBASE, and Capsule-LPI, but also through techniques such as RNA-Seq [72,73,74,75]. The expression profiles of these predicted partners in CRC can then be examined through Western blots, dual luciferase assay, and RNA immunoprecipitation/RNA pull down [76,77]. A more extensive list of methods for the evaluation of lncRNA-dependent phenotypes is provided in Table 1.

## 4. Validated lncRNA Candidates in CRC

Countless studies have identified lncRNAs as candidate biomarkers in CRC. However, many of these studies fail to properly evaluate the role and diagnostic or prognostic potential of these lncRNAs. Carrying out experimental validation of candidate lncRNAs is important to elucidate which candidates are the most promising for future study. A recent study that surveyed a large dataset of lncRNAs found a total of 229 lncRNAs that were differentially expressed in CRC tumors, underscoring CRC heterogeneity. Among the well-established molecular features, the authors classified the tumors based on (i) microsatellite instability (MSI) caused by a deficient DNA mismatch repair (MMR), (ii) chromosome instability (CIN), (iii) the CpG island methylator phenotype (CIMP), and (iv) KRAS, BRAF, and TP53 gene mutations [22]. We surveyed recent studies to compile a list of candidate lncRNA biomarkers (listed in Appendix A) that have been validated in CRC, using at least two of the following four methods: expression profiling, clinical or prognostic significance testing, regulatory significance testing (in vitro), and regulatory significance testing (in vivo). LncRNA candidates were primarily sourced using the cancer lncRNA census 2 (CLC2) [79]. Additionally, we included lncRNAs identified in our previous review on this topic, as well as from a previously published RNA-Seq dataset [23,29]. Testing information for each lncRNA was manually collected by reviewing articles on PubMed. This list includes 117 candidate transcripts, comprising both oncogenic and tumor suppressive lncRNAs and is organized in 4 categories by the extent of reproducibility and experimental characterization. To illustrate the nature of these candidates, we describe some of them here, focusing on two examples of lncRNAs with opposing contributions to CRC and with different levels of characterization, the lesser characterized *LINC01296* and the extensively studied oncogene *HOTAIR*. Additionally, in Figure 1, we provide a comprehensive overview of the most frequently reported mechanism of action of candidate lncRNAs biomarkers, with strong supportive evidence in CRC (as listed in Appendix A).

### 4.1. LINC01296

Long intergenic non-coding RNA 1296, also known as *DUXAP9* (double homeobox A pseudogene 9), is a 68 kb long pseudogene located in 14q11.2. It is associated with cancer, including CRC, and has been recently proposed as a promising biomarker for CRC prognosis by several authors [80]. Information on this gene available at the Genecards website suggests that *LINC01296* is poorly conserved (there are no described orthologs or paralogs), is located in a highly variable region, and is moderately expressed in only six human tissues, which are as follows: white blood cells, lymph nodes, heart, skeletal muscle, adipocyte and kidney [81]. Several studies have explored the expression of *LINC01296* in CRC tissues, cell lines (SW480, SW620, LoVo, HT29, DLD1, HCT116, HCT-8, HCT-8/5-FU) and in vivo rodent models, and have shown overexpression of this lncRNA in tumoral samples compared with matching adjacent tissues [80,82,83]. Additionally, these same studies have shown a significant association with the overexpression of *LINC01296* and tumor stage, lymph node metastasis, distant metastasis, as well as poor clinical prognosis and chemoresistance. In contrast, an earlier meta-analysis found a correlation between *LINC01296* overexpression and better CRC prognosis [84]. Such contradictory results underscore the need for further studies using larger cohorts to elucidate *LINC01296* implications in CRC. In vitro studies have shown the association of *LINC01296* overexpression with cell proliferation, metastasis and chemoresistance to 5-fluorouracil (5-FU), supporting an oncogenic role [82]. Mechanistically, the *LINC01296* transcript functions as an endogenous sponge of miR-26a to regulate mucin1 (MUC1) expression, catalyzed by *GALNT3*, which in turn modulates the activity of the PI3K/AKT pathway, and thereby the carcinogenesis process [82]. Proposed mechanisms of action include down-regulation of p15 for proliferation, and promotion of invasion by regulating the EMT process through the miR-141-3p/ZEB1-ZEB2 axis [80,83].

### 4.2. HOTAIR

The homeobox (HOX) transcript antisense intergenic RNA gene, first described in 2007 [85], is located at the HOXC locus (12q13.13) and encodes for a ~2.4 kb long transcript known as *HOTAIR* (http:/ensembl.org; accessed on 5 April 2022). *HOTAIR* is an oncogenic long intergenic non-coding transcript (lincRNA) that is highly expressed in CRC and multiple other tumor types [86]. *HOTAIR* shows conserved gene structure, high polymorphism (more than 6500 variant alleles have been described (http:/ensembl.org: accessed on 5 April 2022), and low sequence conservation [87]. *HOTAIR* was the first lncRNA described to act in trans [87], as a molecular scaffold, for the assembly of regulatory complexes, including PRC2 and LSD1 proteins, promoting epigenetic repression of the *HOXD* gene, through histone H3K27 trimethylation and histone H3K4me2 demethylation, respectively [85,88]. Other evidence showed a critical effect of *HOTAIR* activity on cell cycle progression and proliferation by regulating different molecules. Recently, novel mechanisms of action, including competitive endogenous RNA (ceRNA) and the microRNA sponge, have also been described [89] (see Figure 1 for further details). Of note, the overrepresentation of sponge and ceRNAs mechanisms of action as the most common for lncRNAs associated with CRC begs the question of whether this is due to study biases—i.e., recent studies looking for lncRNAs that target miRNA, etc.—or whether this is a recurrent mechanism in the context of the disease.

*HOTAIR* expression is positively correlated with the onset and progression of different types of cancer, such as breast, bladder, gastric and CRC, among others [90,91,92]. This overexpression correlates with poor overall survival rate, tumor stage, and metastasis [12]. In CRC, *HOTAIR* causes cell proliferation and metastasis by inducing EMT [93,94]. Several genetic variants have been associated with a high risk of CRC through increased *HOTAIR* overexpression [95,96,97]. In addition, *HOTAIR* transcripts have been detected in heterogeneous types of samples, including serum and other body fluids. *HOTAIR* expression has been shown to be high in CRC cell lines, tumor samples, and metastatic samples [94,98]. Interestingly, a recent study consistently found that downregulation of *HOTAIR* repressed the viability and metastasis of CRC cell lines in vitro, and suppressed the tumorigenesis, migration and invasion of CRC in vivo [94]. Importantly, another study has demonstrated that *HOTAIR* overexpression is associated with resistance to 5-FU-based chemotherapy. Both studies suggest a critical role of this molecule in CRC pathophysiology and a promising role as therapeutic target for CRC patients [98].

Overall, compelling evidence points to a potential role of *HOTAIR* as a biomarker for CRC prediction and diagnosis, due to its detection in bodily fluids [99,100], prognosis [101], and as a potential therapeutic target for the development of novel strategies to target its overexpression [94,98]. So far, a patent has been filed for *HOTAIR* in gastric cancer diagnosis [12], encouraging the development of novel applications.

### 4.3. Other Promising Candidates

Beyond *LINC01296* and *HOTAIR*, countless other candidates have accumulated increasing evidence of a strong regulatory role in CRC. Some of the most well studied lncRNA candidates include the oncogenes *NEAT1* and *CCAT1* and the tumor suppressor genes *GAS5* and *MEG3* (Table 2). Notably, these lncRNA candidates often have additional, sometimes opposing, roles in other types of cancer. This emphasizes the need for tissue-specific studies to accurately evaluate the therapeutic potential of these lncRNAs. As lncRNA candidates have become more widely studied, several controversies over their role within a cancer specific context have ensued. Two of these controversies surround the lncRNAs *MALAT1* and *BANCR* in CRC. Initially, these lncRNAs were thought to be overexpressed in CRC tissues, and therefore were proposed as oncogenic factors that promoted tumorigenesis [102]. In the case of *BANCR*, initial findings suggested that it induced EMT through a MEK signaling pathway, ultimately leading to an increase in the migratory abilities of cancer cells [102]. In the case of *MALAT1*, initial studies observed several mutations at the 3′ end of the *MALAT1* transcript, elucidating its role in promoting cell migration and invasion in CRC [103]. While numerous studies have corroborated findings in support of the oncogenic role of *BANCR* and *MALAT1*, others have challenged these claims. In direct conflict with the initial studies, researchers using an in vivo model of *BANCR* overexpression in CRC found that the overexpression of *BANCR* inhibits tumor growth [104]. Conversely, another group using a similar in vivo model found that silencing of *BANCR* inhibits tumor growth, in line with the findings of initial studies [105]. Similarly, while *MALAT1* has been extensively studied as an oncogenic lncRNA, a recent study suggests that downregulation of *MALAT1* results in increased cancer cell migration and correlates with reduced patient survival [106]. These findings suggest that *MALAT1* acts as a tumor suppressor, contrary to previous evidence [106,107]. These two examples illustrate the complexity of lncRNAs roles in cancer and underscore the need for further studies to clarify their roles and their potential as diagnostic or prognostic markers or as targets for new therapies.

## 5. Current Use of lncRNAs in Clinical Practice

LncRNAs are actively being explored as independent biomarkers or in combination with other lncRNAs or proteins for diagnosis and prognosis. The first lncRNA to be validated as a biomarker and used in clinical practice was *PCA3* for the diagnosis of prostate cancer [11]. More recently, several clinical trials have investigated the association of lncRNAs with several diseases (for example, preeclampsia, NCt03903393; lung cancer, NCt03830619; acute ischemic stroke, NCt04175691) (Winkle et al., 2021). For CRC, there are two clinical trials underway at the time of writing this review [108]. The first explores the application of *CCAT1* for early diagnosis of the disease (NCT04269746). The second examines the role of *HOTTIP* (in combination with the protein EIF4EBP1) as a biomarker for diagnosis and prognosis by measuring its abundance in serum and the presence of the single nucleotide polymorphism *HOTTIP* rs1859168 (NCT04729855). Both lncRNAs passed our selection criteria and are included in Appendix A. Importantly, due to the efforts over the last decade, the list of lncRNA candidates for pre-clinical and clinical studies is growing, (see Appendix A), which has resulted in more than 4000 patents for lncRNAs in CRC [109].

RNA-based drugs target the proteome, the transcriptome (including non-coding RNA), and the genome in an unprecedented way [110]. RNA therapeutics are specific and are able to target both common and rare or untargetable diseases in a personalized manner [111]. Nowadays, there is a growing interest in novel treatment strategies, with special focus on lncRNAs due to their advantageous characteristics, such as tissue specificity, regulatory roles, aberrant expression in cancer, and being detectable in serum samples. Additionally, at least three small molecules have been proven to prevent the action of two lncRNAs, *GAS5* and *MALAT1*, related with diabetes and carcinogenesis, respectively [112]. Two clinical studies (NCT02508441; NCT03985072) have tested the effects of ASOs targeting mitochondrial lncRNA on solid tumors [113]. Overall, lncRNAs represent a promising source of novel tools in the RNA-therapeutics era that can be applied to a plethora of conditions, including CRC.

## 6. Conclusions

CRC continues to be one of the most common cancers worldwide, affecting millions of people each year. With an increasingly younger patient cohort and a high mortality rate, there is great need for the development of less invasive and more timely techniques for the diagnosis, prognosis and treatment of CRC. Previously, non-invasive diagnostic techniques and prognostic monitoring of CRC have been limited. However, recent studies have focused on novel tools and strategies for improving both our understanding of the disease and the perspective of CRC patients. In this review, we highlight the study of lncRNAs as potential biomarkers for the diagnosis, prognosis, and treatment of CRC, including an overview of the most commonly implemented methods for the detection and validation of these lncRNAs. While much of the research has focused on the discovery of these candidate biomarkers, it is paramount that lncRNA candidates are validated at the molecular, prognostic, and regulatory level. In this way, future research should focus on the most therapeutically actionable lncRNA targets. Although lncRNAs are not yet a common tool in CRC clinical practice, their growing use in other cancers is encouraging for the application of RNA therapeutics and personalized medicine in CRC. In the future, the continued evaluation of candidate biomarkers and subsequent clinical trials will undoubtedly aid in the transition to more personalized approaches for assessing CRC cases. By improving diagnosis, prognostic predictions, and targeted therapies, lncRNAs have the potential to alter treatment strategies and clinical outcomes for the better.

## Figures and Tables

**Figure 1 cancers-14-03866-f001:**
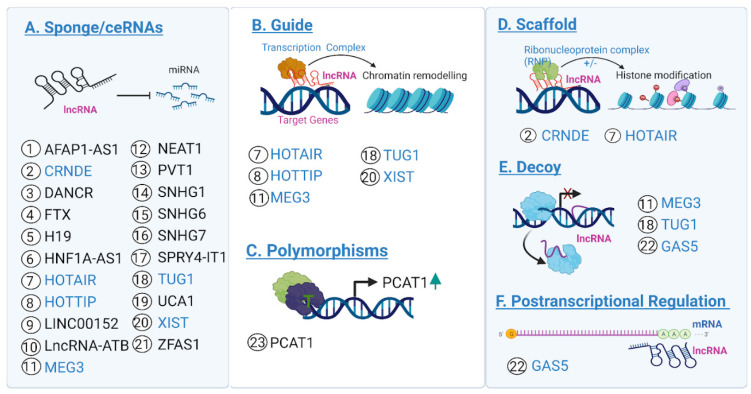
lncRNAs in CRC. Schematic description of the most frequent mechanisms of action (from **A**–**F**) reported for the list of the candidate lncRNAs biomarkers (indicated by numbered circles) listed in Table 2, with strong supporting evidence in CRC. Highlighted in blue are the names of the lncRNAs with more than one mechanism reported. Created using Biorender.com.

**Figure 2 cancers-14-03866-f002:**
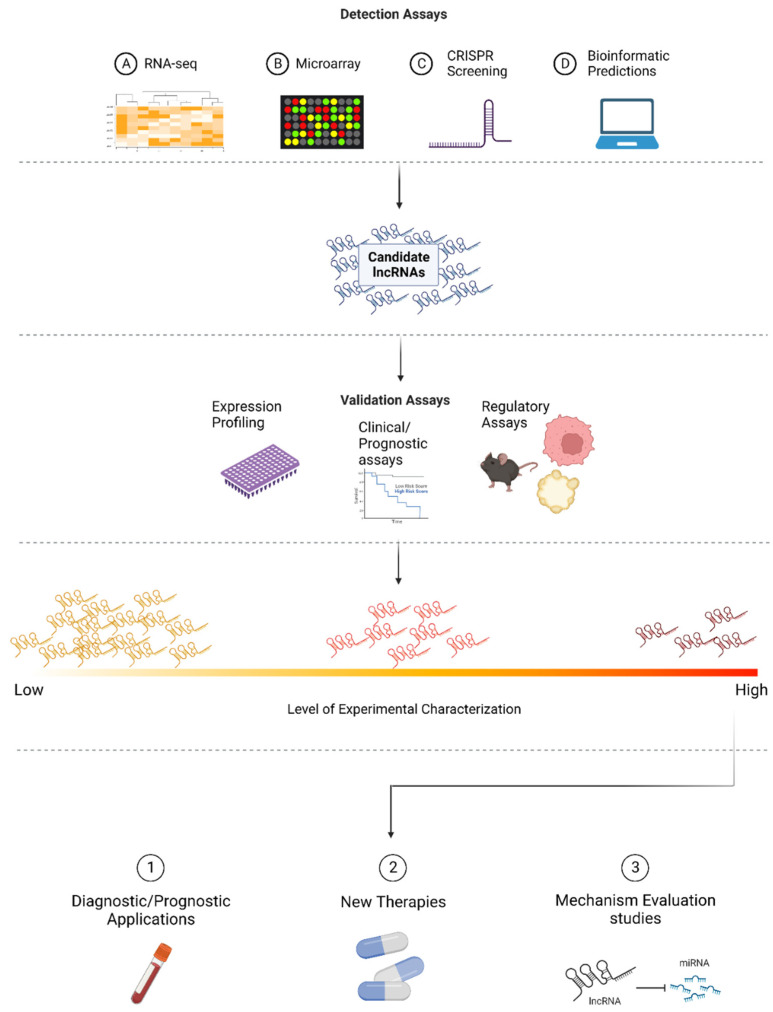
Methods for detecting and validating lncRNA biomarkers in CRC. After detection of candidate lncRNAs via the methods shown above (see text for full descriptions), validation techniques are used to characterize and assess which candidate lncRNAs are the most suitable for future studies. Regulatory assays may be carried out in cell lines, organoids, or in vivo models. Well characterized lncRNAs can be used as diagnostic/prognostic tools, as targets in future therapies, and as subjects of mechanistic studies. Created with Biorender.com.

**Figure 3 cancers-14-03866-f003:**
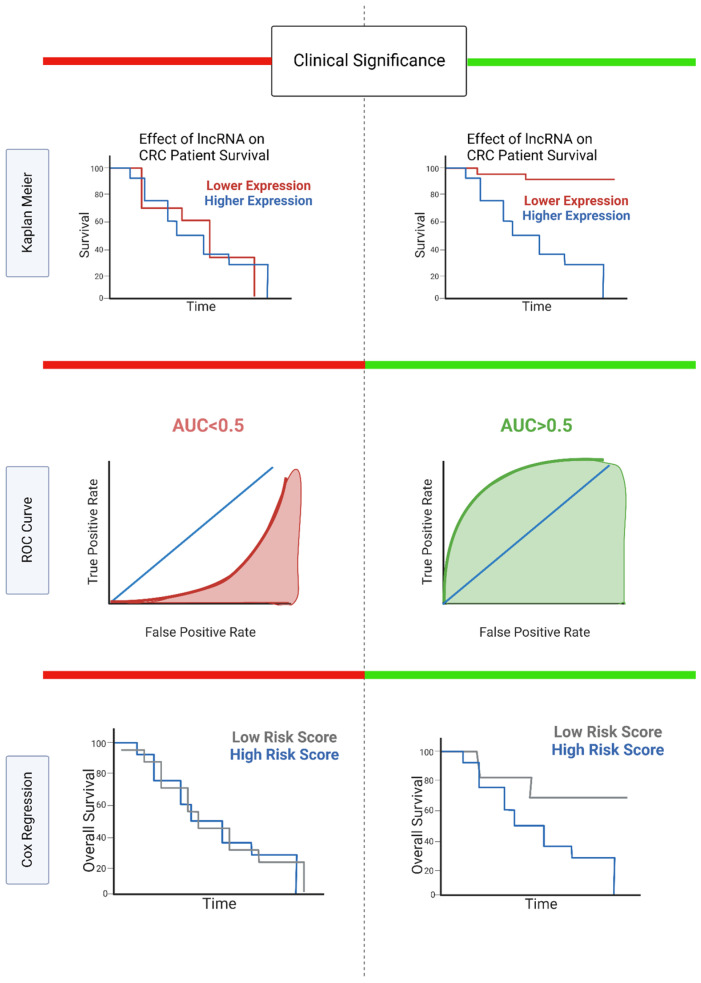
Evaluating clinical and prognostic significance of lncRNAs in CRC. Both insignificant (**left**, **in red**) and significant (**right**, **in green**) example graphics for Kaplan–Meier survival analysis, ROC curve analysis, and Cox regression analysis are shown. Created using Biorender.com.

**Table 1 cancers-14-03866-t001:** Methods for determining lncRNA-dependent phenotypes of CRC.

Methodology	Application	Model	Biological Process Tested	Dysregulation of lncRNA of Interest Required	Refs. (PubMed ID) ^1^
Apoptotic assay	In vitro/in vivo	Cell lines, organoids, animal assays	Cell proliferation	Yes	32144238
Transwell assay	In vitro	Cell lines, organoids	Cell invasion	Yes	33328585
CCK-8 assay	In vitro	Cell lines, organoids	Cell proliferation	Yes	34224294
MTT assay	In vitro	Cell lines, organoids	Cell proliferation	Yes	33277833
Wound healing assay	In vitro	Cell lines, organoids	Cell migration	Yes	33570445
Colony formation assay	In vitro	Cell lines	Cell formation	Yes	34371180
Flow cytometry	In vitro	Cell lines	Cell cycle/apoptosis	Yes	33099922
Bioinformatic programs (RPIseq, lncPRO, lncBASE, Capsule-LPI)	In silico	N/A	Coexpression networking	No	35034547
RNA sequencing	In vitro	Cell lines, organoids	Coexpression networking	Yes	35039060
Western blot	In vitro/in vivo	Cell lines, organoids, animal assays	Protein expression	Yes	34498706
Dual luciferase assay	In vitro	Cell lines	Interactions	Yes	35066433
RNA immunoprecipitation	In vivo	Cell lines, organoids, animal assays	Interactions	Yes	35110535
RNA pull-down	In vitro	Cell lines, organoids	Interactions	Yes	35107754
Tumor formation assay	In vivo	Organoids, animal assays	Tumor formation	Yes	34477476

^1^ PMID = PubMed ID number [78].

**Table 2 cancers-14-03866-t002:** LncRNA candidates with strong supporting evidence for a regulatory role in CRC.

lncRNA	Mechanism of Action	Refs. (Pubmed ID) ^1^
** *AFAP1-AS1* **	Proliferation, migration, invasion through the miR-195-5p/WISP1 axis. Tumor growth and metastasis	34335760, 27578191
** *CRNDE* **	Regulation of apoptosis, proliferation, drug sensitivity via the Akt/mTORC1 pathway. Epigenetic transcriptional regulation of *DUSP5* and *CDNK1A*	35069879, 28796262
** *DANCR* **	Suppression of apoptosis via RNA stabilization of *MALAT1* Enhanced growth and metastasis via the *DANCR*/miR-518a-3p/MDM2 ceRNA network	33414433, 32423468
** *FTX* **	Proliferation, migration, invasion through the *FTX*-miR-214-5p-JAG1 regulatory axis. Enhanced growth and progression via the miR-192-5p/EIF5A2 axis	34733921, 32280242
** *GAS5* **	Inhibition of proliferation and migration, induction of apoptosis via the *GAS5*/miR-10b axis. Suppression of macroautophagy, induction of apoptosis via the mTOR/SIRT1 pathway	35103069, 33416133
** *H19* **	Migration, invasion, induction of EMT, metastasis via activation of Raf-ERK signaling Proliferation, invasion, metastasis via the *H19*/miR-29b-3p/*PGRN*/Wnt axis	32698890, 29754471
** *HNF1A-AS1* **	Migration, invasion, glycolysis via miR-124/MYO6. Angiogenesis via the PBX3/OTX1/ERK-MAPK pathway	32110048, 32325080
** *HOTAIR* **	Migration, invasion, EMT, cell viability via SNAIL/HNF4α transcriptional regulation. Suppression of miR-218 via the EZH2-targeting miR-218-2 promoter regulatory axis	33588137, 28918035
** *HOTTIP* **	Proliferation, migration, invasion Enhanced susceptibility via rs3807598, rs2067087, and rs17427960 SNPs	31945724, 30940774
** *LINC00152* **	Proliferation and metastasis via promoter hypomethylation and the YAP1*/LINC00152*/miR-632/miR-185-3p/FSCN1 axis	32307642, 32042551
** *lncRNA-ATB* **	Proliferation, migration, invasion via sponging miR-141-3p, metastasis Developmental flexibility via transcriptional regulation of β-catenin	33199986, 32256798
** *MEG3* **	Inhibited proliferation through targeting SOCS3/STAT3 signaling via miR-708 Inhibited proliferation and migration via the miR-376/PRDK1 signal axis	34934045, 31632544
** *NEAT1* **	Proliferation, invasion, apoptotic suppression via the miR-138/*SLC38A1* axis Proliferation via the KDM5A/Cul4A/Wnt axis	32700988, 34109988
** *PCAT1* **	Proliferation, migration, invasion, drug resistance. Proliferation, migration, invasion, apoptotic suppression via miR-149-5p regulation	33277833, 31646561
** *PVT1* **	Proliferation, apoptotic regulation via the miR-761/MAPK1 axis. Epigenetic regulation of MYC, regulation of TGFβ/SMAD and Wnt/β-Catenin pathways	34515320, 33148262
** *SNHG1* **	EMT regulation via miR-497-5p/miR-195-5p modulation. Proliferation, migration, invasion via Wnt/β-catenin signaling	31276207, 29749530
** *SPRY4-IT1* **	Cell growth and glycolysis via PDK1. Proliferation, migration, invasion, EMT regulation via miR-101-3p modulation	33029299, 28720069
** *TUG1* **	Proliferation, invasion, migration, apoptotic suppression, tumor growth via the miR-542-3p/TRIB2 axis and Wnt/β-catenin pathway. Proliferation, migration, cell viability via the *TUG1*/miR-145-5p/TRPC6 regulatory axis	34030715, 32985219
** *TUSC7* **	Inhibition of proliferation, invasion, EMT, enhanced apoptosis via the *TUSC7*/miR-23b/PDE7A axis	33370523, 31002365
** *UCA1* **	Proliferation, migration, invasion, EMT, drug resistance via the *UCA1*/miR-495-SP1/SP3 axis. Proliferation and drug resistance via *UCA1*/miR-495-HGF/c-MET	33961855, 34976187
** *XIST* **	Proliferation, EMT, drug resistance via the *XIST*/miR-125b-2-3p/WEE1 axis. Proliferation, migration, invasion, apoptotic suppression via the miR-338-3p/PAX5 axis	33666372, 32826710
** *ZEB1-AS1* **	Proliferation via miR-141-3p regulation. Cell viability and apoptotic suppression via the MiR-205/YAP1 axis	32669962, 32190742
** *ZFAS1* **	Tumor size, metastasis, lipogenesis via PABP2/SREBP1. Proliferation, migration, invasion, metastasis via miR-34b/SOX4	35036050, 33725330
** *SNHG6* **	Proliferation, apoptotic suppression via JAK2/*SNHG6* regulation. Proliferation and invasion via miR-101-3p regulation and the Wnt/β-catenin signaling pathway	32840014, 31533634
** *CCAT2* **	Proliferation, apoptotic suppression. Proliferation, migration, invasion via TAF15/RAB14/AKT/GSK3β axis, tumor growth and metastasis	33099922, 34868956
** *SNHG7* **	Proliferation, migration, invasion, cell viability, and metastasis via miR-216b regulation and *GALNT1* expression	29915311, 33685194
** *FOXD2-AS1* **	Proliferation, cell cycle regulation via miR-4306 regulation. Proliferation, migration, invasion via the miR-25-3p/Sema4C axis	34396433, 31908535
** *LINC00460* **	Metastasis via miR-149-5p and biglycan regulation. Proliferation, migration, invasion, apoptotic suppression via the miR-613/SphK1 axis	33472555, 32821121
** *MIR4435-2HG* **	Proliferation, migration, invasion, metastasis via the miR-206/YAP1 axis. Proliferation, apoptotic suppression	32154166, 32141545
** *ELFN1-AS1* **	Proliferation, migration, invasion, apoptotic suppression via the miR-1205/MTA1 axis. Proliferation, migration, apoptotic suppression via the miR-4644/TRIM44 axis	34337713, 31929141
** *LINC00858* **	Suppression of apoptosis, senescence, autophagy. Tumor growth via WNK2 regulation. Proliferation, invasion, migration via the miR-4766-5p/PAK2 axis	32768499, 31902050
** *CCAT1* **	Proliferation, migration, invasion via the hsa-miR-4679/GNG10 axis. Migration, invasion, cell viability via the *CCAT1*/VEGF/miR-218 axis	35005034, 32256733

^1^ PMID = PubMed ID number [78].

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
