# Peer review of "Discovery and Validation of Clinically Relevant Long Non-Coding RNAs in Colorectal Cancer"

_cancers, 2022, doi:10.3390/cancers14163866_

Round 1

Reviewer 1 Report

Snyder et al provide a comprehensive overview of the main approaches for discovering and validating lncRNAs in colorectal cancer (CRC) and the list of lncRNAs associated with CRC. The authors focused on different technologies, disease and have provided few examples of lncRNAs that have been implicated in CRC. To distinguish this review from other reviews on the same topic, this manuscript would be significantly strengthened and improved by addressing the points outlined below:

MAJOR COMMENTS

·       Figure 1 and in general the manuscript would benefit if the authors mentioned cellular models used to study CRC: from cancer colorectal cell lines to well established colon organoids. In Figure 1 in addition to mouse, the cells and organoids could be added as alternative model to study the function of lncRNAs in CRC. 

·       Since the review is on lncRNAs, I was expecting more information about lncRNA biology and their roles in cancer. For example, how is lncRNA expression deregulated in cancer, what are the mechanisms?  How are lncRNAs altered in CRC and what is the contribution of CRC drivers in regulation of lncRNA expression in CRC.

·       Molecular  features of CRC are well established from (i) microsatellite instability (MSI) caused by a deficient DNA mismatch repair (MMR), (ii) chromosome instability (CIN), (iii) the CpG island methylator phenotype (CIMP), and (iv) KRAS, BRAF, and TP53 gene mutations. Although many lncRNAs such MALAT1, HOTAIR and CCAT1 have been implicated in CRC it is not clear in which type of CRC lncRNAs  could intervene. For example, the authors should mention if there is any group of lncRNAs that is known to correlate with the specific genomic-related characteristic of CRC (eg MMR status or CIN low or CIN High). Please see PMID: 29963224

·       How many lncRNA genes have differential expression based on MMR, CIMP or CIN status or mutations in CRC drivers? 

·       The authors mentioned OncodriverFML paper as a potential platform to look into lncRNA mutations in CRC. In general how many lncRNAs have been shown to be mutated in cancer and specifically in CRC? The paper mentioned MALAT1 and MIAT1 but PCAWG studies showed that although lncRNAs NEAT1 and MALAT1 are mutated in cancer, these mutations are most likely due to transcription-associated mutational process (PMID: 32025015).

·       The authors should also include lncRNAs with a potential role in colorectal adenoma-carcinoma transition, please see PMID: 31694571.

  • Are there any CRC specific lncRNAs such as SAMMSON, a melanoma specific lncRNA or all mentioned lncRNAs are expressed in many types of cancer incl. CRC?

·       In addition to RNA FISH and expression profiling, RNA scope is a powerful method to analyse the expression of lncRNAs in cancer tissues.

·       In addition to several bioinformatics techniques to identify lncRNA targets, the authors should mention also RNA atlas from Lorenzi et al 2021; PMID: 34183863. This platform could also be useful for CRC.

·       Table 1 could also include not only  the methods but also cellular models to study lncRNAs: CRC cell lines (eg CIN + and CIN-) , colon organoids and APC mice models. 

MINOR COMMENTS

Typo error in the abstract, it should be long noncoding RNAs not “long, noncoding RNAs”. Please check is it lncRNAs or lncRNA.

Author Response

REVIEWER 1

Snyder et al provide a comprehensive overview of the main approaches for discovering and validating lncRNAs in colorectal cancer (CRC) and the list of lncRNAs associated with CRC. The authors focused on different technologies, disease and have provided few examples of lncRNAs that have been implicated in CRC. To distinguish this review from other reviews on the same topic, this manuscript would be significantly strengthened and improved by addressing the points outlined below:

MAJOR COMMENTS

  •       Figure 1 and in general the manuscript would benefit if the authors mentioned cellular models used to study CRC: from cancer colorectal cell lines to well established colon organoids. In Figure 1 in addition to mouse, the cells and organoids could be added as alternative model to study the function of lncRNAs in CRC. 

Response: Thank you, we have now added some text on cell lines and organoids, and added these models to Figure 1. 

  •       Since the review is on lncRNAs, I was expecting more information about lncRNA biology and their roles in cancer. For example, how is lncRNA expression deregulated in cancer, what are the mechanisms?  How are lncRNAs altered in CRC and what is the contribution of CRC drivers in regulation of lncRNA expression in CRC.

Response: Thank you, we have added a new paragraph on lncRNA biology and their mechanism in cancer.

  •       Molecular  features of CRC are well established from (i) microsatellite instability (MSI) caused by a deficient DNA mismatch repair (MMR), (ii) chromosome instability (CIN), (iii) the CpG island methylator phenotype (CIMP), and (iv) KRAS, BRAF, and TP53 gene mutations. Although many lncRNAs such MALAT1, HOTAIR and CCAT1 have been implicated in CRC it is not clear in which type of CRC lncRNAs could intervene. For example, the authors should mention if there is any group of lncRNAs that is known to correlate with the specific genomic-related characteristic of CRC (eg MMR status or CIN low or CIN High). Please see PMID: 29963224. How many lncRNA genes have differential expression based on MMR, CIMP or CIN status or mutations in CRC drivers?

Response: Thank you, we have revised the literature and confirmed that as suggested by de Bony et al., (PMID:29963224), there is a group of 282 lncRNAs whose expression reflects CRC heterogeneity. We included this information in the main text· 

  •       The authors mentioned OncodriverFML paper as a potential platform to look into lncRNA mutations in CRC. In general how many lncRNAs have been shown to be mutated in cancer and specifically in CRC? The paper mentioned MALAT1 and MIAT1 but PCAWG studies showed that although lncRNAs NEAT1 and MALAT1 are mutated in cancer, these mutations are most likely due to transcription-associated mutational process (PMID: 32025015). -

Response: We agree with the reviewer. PCAWG studies showed mutations of lncRNA, for instance NEAT1 and MALAT1 associated with cancer without causal implication. In our study we focus on clinically relevant molecules altered in cancer. 

  •       The authors should also include lncRNAs with a potential role in colorectal adenoma-carcinoma transition, please see PMID: 31694571.

Response: Thank you for the suggestion. We have added several lncRNAs to our supplementary figure that are mentioned in this paper. Many of the lncRNAs from this paper were already in our table or did not fit the criteria for two or more sources of supporting evidence. Those that could be added have been added here. 

  • Are there any CRC specific lncRNAs such as SAMMSON, a melanoma specific lncRNA or all mentioned lncRNAs are expressed in many types of cancer incl. CRC?

Response: We thank the reviewer for their question. In the supplementary figure, lncRNAs involved in CRC are categorized based on their role in CRC but also their role in other types of cancer. The column of the table “onco vs. ts other” describes the role each lncRNA plays in cancers other than CRC. For those lncRNAs specific to CRC, this column displays N/A, as there is no evidence of a role in any other type of cancer. 

  •       In addition to RNA FISH and expression profiling, RNA scope is a powerful method to analyse the expression of lncRNAs in cancer tissues.

Response: We have now added this technique. 

  •       In addition to several bioinformatics techniques to identify lncRNA targets, the authors should also mention RNA atlas from Lorenzi et al 2021; PMID: 34183863. This platform could also be useful for CRC.

Response: We thank the reviewer for the comment. We have now added this source of lncRNA targets 

  •       Table 1 could also include not only  the methods but also cellular models to study lncRNAs: CRC cell lines (eg CIN + and CIN-) , colon organoids and APC mice models. 

Response: We now refer to which methods could be applied in organoids,, cell lines or mice models, making an exhaustive list of models is beyond the scope of this manuscript. 

MINOR COMMENTS 

Typo error in the abstract, it should be long noncoding RNAs not “long, noncoding RNAs”. Please check is it lncRNAs or lncRNA.

Response: This has been corrected.

Reviewer 2 Report

In this review, the authors evaluate historical and current scientific advances in lnRNA role in colorectal cancers. In the last couple of decades, lnRNAs have emerged as important regulators of biological processes, particularly transcription and translation. Additionally, they show differential expression in various cells. Changes in expression also occurs in cancer cells; hence their use as biomarkers.

This review is timely and well written. The authors provide a short historical view but pay particular attention to recent discoveries. Further, and perhaps importantly, they highlight certain key lnRNAs in two tables in the main text and one supplemental and provide references from literature. This reviewer appreciates this because should the reader want more information, they can go to the references provided.

In this reviewer's opinion, this report should be published in Cancers. I have no major comments!

The only minor comment I have is to re-format the two tables so that PubMed ID is given as reference in both. At the moment, table 1 has a classical way of listing references (e.g. Y. Liu et al.2020) while table 2 gives pubmed IDs. Also, in table 2 there is no reason to write PMID for every reference. I would just change the reference heading to "References (PubMed ID)"  Then in the footnote, you can describe in some more detail what than number is perhaps by also including the PubMed link (https://pubmed.ncbi.nlm.nih.gov/) in the unlikely case that someone uses google scholar and never heard of pubmed!!

Author Response

REVIEWER 2

In this review, the authors evaluate historical and current scientific advances in lnRNA role in colorectal cancers. In the last couple of decades, lnRNAs have emerged as important regulators of biological processes, particularly transcription and translation. Additionally, they show differential expression in various cells. Changes in expression also occurs in cancer cells; hence their use as biomarkers.

This review is timely and well written. The authors provide a short historical view but pay particular attention to recent discoveries. Further, and perhaps importantly, they highlight certain key lnRNAs in two tables in the main text and one supplemental and provide references from literature. This reviewer appreciates this because should the reader want more information, they can go to the references provided.

In this reviewer's opinion, this report should be published in Cancers. I have no major comments!

Response: We thank reviewer 2 for his/her nice comments and appreciation of our work.  

The only minor comment I have is to re-format the two tables so that PubMed ID is given as reference in both. At the moment, table 1 has a classical way of listing references (e.g. Y. Liu et al.2020) while table 2 gives pubmed IDs. Also, in table 2 there is no reason to write PMID for every reference. I would just change the reference heading to "References (PubMed ID)"  Then in the footnote, you can describe in some more detail what than number is perhaps by also including the PubMed link (https://pubmed.ncbi.nlm.nih.gov/) in the unlikely case that someone uses google scholar and never heard of pubmed!!

Response: We have changed the heading of table 2 and explained in the legend what is the pubmed link (and provided the URL)